



# Spatial and temporal representativeness of point measurements for nitrogen dioxide pollution levels in cities

Ying Zhu[a,b], Jia Chen[b], Xiao Bi[b], Gerrit Kuhlmann[c], Ka Lok Chan[d], Florian Dietrich[b], Dominik Brunner[c], Sheng Ye[a], and Mark Wenig[a]

[a]Meteorological Institute, Ludwig-Maximilians-Universität München, Munich, Germany
[b]TUM Department of Electrical and Computer Engineering, Technische Universität München, Munich, Germany
[c]Empa, Swiss Federal Laboratories for Materials Science and Technology, Überlandstrasse 129, Dübendorf, Switzerland
[d]Remote Sensing Technology Institute (IMF), German Aerospace Center (DLR), Oberpfaffenhofen, Germany

**Correspondence:** Y.Zhu (ying.zhu@physik.uni-muenchen.de), M.Wenig (mark.wenig@lmu.de), J.Chen (jia.chen@tum.de)

**Abstract.** In many cities around the world the overall air quality is improving, but at the same time nitrogen dioxide ($NO_2$) trends show stagnating values and in many cases could not be reduced below air quality standards recommended by the World Health Organization (WHO). Many large cities have built monitoring stations to continuously measure different air pollutants. While most stations follow defined rules in terms of measurement height and distance to traffic emissions, the question remains, how representative are those point measurements for the city-wide air quality. The question of the spatial coverage of a point measurement is important because it defines the area of influence and coverage of monitoring networks, determines how to assimilate monitoring data into model simulations or compare to satellite data with a coarser resolution, and is essential to assess the impact of the acquired data on public health.

In order to answer this question, we combined different measurement data sets consisting of path averaging remote sensing data and in-situ point measurements in stationary and mobile setups from a measurement campaign that took place in Munich, Germany in June and July 2016. We developed an algorithm to strip temporal diversity and spatial patterns, in order to construct a consistent $NO_2$ pollution map for Munich. Continuous long-path differential optical absorption spectroscopy (LP DOAS) measurements were complemented with mobile cavity-enhanced (CE) DOAS, chemiluminescence (CL) and cavity attenuated phase shift (CAPS) instruments and were compared to monitoring stations and satellite data. In order to generate a consistent composite map, the LP DOAS diurnal cycle has been used to normalize for the time of the day dependency of the source patterns, so that spatial and temporal patterns can be analyzed separately. The resulting concentration map visualizes pollution hot spots at traffic junctions and tunnel exits in Munich, providing insights into the strong spatial variations. On the other hand, this database is beneficial to the urban planning and the design of control measures of environment pollution. Directly comparing on-street mobile measurements in the vicinity of monitoring stations resulted in a difference of $48\%$. For the extrapolation of the monitoring station data to street level, we determined the influence of the measuring height and distance to the street. We found that a measuring height of $4\,m$, at which the Munich monitoring stations measure, results in $16\%$ lower average concentrations than a measuring height of $1.5\,m$, which is the height of the inlet of our mobile measurements and a typical pedestrian breathing height. The horizontal distance of most stations to the center of the street of about $6\,m$ also results in an average reduction of $13\%$ compared to street level concentration. A difference of $21\%$ in the $NO_2$ concentrations





remained, which could be an indication that city-wide measurements are needed for capturing the full range and variability of concentrations for assessing pollutant exposure and air quality in cities.

# 1 Introduction

Many former studies (Huang et al., 2014; Dunlea et al., 2007; Jang and Kamens, 2001) have been pointed out that $NO_2$ is an important composition in the process of both tropospheric and stratospheric chemistry. It is one of the major pollution products from combustion processes. Catalytic formation of tropospheric ozone ($O_3$) and the formation of secondary aerosols that cause acid rain, all of which involve its participation. Elevated concentration of atmospheric $NO_2$ is acknowledged to be noxious to human beings health. In urban environments, Exhaust emissions are one of the primary sources of air pollution, particularly $NO_x$ (= NO + $NO_2$). Nitrogen-monoxide (NO) accounts for the majority of direct traffic emissions, which is subsequently oxidized to form $NO_2$ although some $NO_2$ is emitted directly (Ban-Weiss et al., 2008; Henderson et al., 2007; Kirchstetter et al., 1999). $NO_2$ levels are often strongly correlated with many other toxic air pollutants. Its concentration can be easily and precisely measured, which is helpful in assessing general air quality. Since it is a short-lived compound gas from numerous different sources, its concentrations can vary strongly, both in space and time.

According to the 2017 European Environment Agency report (EEA, 2017), some $NO_2$ concentrations measured at air quality monitoring stations are above the World Health Organization (WHO) Air Quality Guideline (AQG) values of $200\,\mu g/m^3$ (hourly) and $40\,\mu g/m^3$ (annually). $10.5\,\%$ of the stations across European cities exceeded the annual limits including several German cities. None of the exceedances were observed at rural background stations, but in urban or suburban stations. More specifically, $89\,\%$ of the exceeded values were observed at traffic stations. The 2016 air quality report by the German Environment Agency (Umweltbundesamt) (UBA, 2017) also pointed out that the air pollution in urban conurbations was primarily affected by traffic. In the 2015 technical report by the Bavarian environment agency (Landesamt für Umwelt, LfU) (LfU, 2015), the on-road $NO_2$ concentration limits were exceeded in most Bavarian cities from 2000 to 2014. In particular, the annual $NO_2$ level in Munich measured at Landshuter Allee station was more than twice of the annual $NO_2$ limit value of the WHO AQG of $40\,\mu g/m^3$.

With a growing focus on air pollution in the public attention, stationary monitoring networks have been established all over the world. Monitoring stations continuously measures different pollutants and while most stations follow defined rules in terms of measurement height and distance to traffic emissions, the question remains, how representative are those point measurements for the city wide air quality. According to a study of the spatial distribution of $NO_2$ in Hong Kong (Zhu et al., 2018), large differences between mobile measurements around the city and seven local monitoring stations were observed. In order to determine the representativeness of air quality monitoring stations, different measurement methods have to be combined. Most monitoring stations utilize the ChemiLuminescence (CL) technique for $NO_x$ measurements. Thereby the $NO_2$ concentration is determined indirectly by calculating the difference between $NO_x$ and NO concentrations. The concentration of oxidized odd-Nitrogen species ($NO_y$) is inevitably included as a small measurement error. Nevertheless, the CL technique has a good detection sensitivity that is given by its low background signal. This is because for initiating the fluorescence no light source





is required (Dunlea et al., 2007). In this study, we compared our CL and cavity-enhanced DOAS (CE DOAS) data to the local air quality stations and studied the diffusion rate of $NO_2$ in both vertical and horizontal directions from one of the stations.

For our study we utilized a combination of long-path DOAS (LP-DOAS) instrument and a CE DOAS, as well as a Cavity Attenuated Phase shift Spectroscopy (CAPS) instrument to determine the spatio-temporal variability of $NO_2$ concentrations

in the central area of Munich, CE DOAS is a spectroscopic measurement technique that uses an optical resonator to fold the absorption path into into a resonator (Zhu et al., 2018; Min et al., 2016; Thalman and Volkamer, 2010; Platt et al., 2009; Washenfelder et al., 2008; Venables et al., 2006; Langridge et al., 2006). CAPS (Herbelin et al., 1980) is a spectroscopic detection technology, generally referred to cavity enhanced optical absorption, which has also been applied for the detection of atmospheric pollutants in many studies (Xie et al., 2019; Kundu et al., 2019; Ge et al., 2013; Kebabian et al., 2008, 2005a).

The advantage of CE DOAS and CAPS is the fact that they are not sensitive to other reactive nitrogen oxides in the atmosphere like some other in-situ $NO_2$ monitoring techniques. They are both characterized by a compact setup and have no sensitivity loss during the operation. For mobile measurements a fast sampling rate is necessary, and the high accuracy of the instruments allowed a sampling rate of $2\,s$. Similar instrument setups have been used in many on-road studies of vehicles emissions (Zhu et al., 2018; Chan et al., 2017; Rakowska et al., 2014; Ning et al., 2012; Uhrner et al., 2007; Vogt et al., 2003).

In order to verify whether the LP DOAS measurements are representative for the whole city, $NO_2$ data from the Ozone Monitoring Instrument (OMI) on-board the NASA Aura satellite was used. Satellite measurements are commonly used for global scale long-term observation of aerosols and trace gases (Silvern et al., 2019; Zara et al., 2018; Laughner and Cohen, 2017; Inness et al., 2015). Validation studies revealed that satellite retrievals generally underestimate urban areas but also found good correlations between satellite and ground based observations (Chan et al., 2018; Lin et al., 2012; Lamsal et al., 2008;

Wenig et al., 2008; Petritoli et al., 2004).

For our study we conducted on-road measurements of $NO_2$ concentrations in June and July of 2016 in order to investigate street level air quality and locate emission hot spot areas. Additionally, LP DOAS measurements were conducted to observe the temporal variability of ambient $NO_2$ in Munich. A measurement system consisting of several DOAS instruments was continuously operational for over 2 years (see Sec. 2). The algorithm that combines the mobile and stationary measurement

data is described in Section 3.1.1. The resulting on-road $NO_2$ spatial patterns are presented in Section 3.1.2. The CE DOAS and CL were set up next to the Bavarian LfU local air quality station to measure the horizontal and vertical $NO_2$ distributions, and the results are shown in Section 3.2. In addition, Section 3.3 presents the comparison of LP DOAS $NO_2$ measurements with OMI satellite data, and analyzes the characteristics of seasonal $NO_2$ variation.

## 2   Methodology

This study combines different measurement methods such as mobile, stationary and satellite measurements to answer the question of how representative sparse point measurements are to determine the air quality of a city. Furthermore, we want to find out what kind of measurement approach is needed to determine the overall air quality in a city. The Munich three-dimensional DOAS measuring system combines three different types of DOAS instruments, specifically, CE DOAS and LP





DOAS. The measurement system is installed on the roof of the building of the Meteorological Institute Munich (MIM) at the Ludwig Maximilians University (LMU) in the center of Munich. The three LP DOAS instruments scan retro reflector arrays in different directions and distances, capturing the horizontal variations at the rooftop level. A CE DOAS is used to determine the $NO_2$ variability on the ground. The LP DOAS instruments run continuously, whereas the CE DOAS is used at different times

of the year/week/day under varying meteorological conditions to determine street-level $NO_2$ distributions.

## 2.1  Mobile Measurements

A CE DOAS and a CAPS instruments in two vehicles were used for on-street sampling of traffic emissions. The sampling inlets were located at the front right window of each vehicle at $1.5\,\mathrm{m}$ height. For measurements in Munich's city park (English Garden), we used a bike trailer. The measurements were performed in June and July 2016 to cover a large part of the urban area

in Munich. The sample resolution of the CE DOAS and the CAPS were both adjusted to $2\,\mathrm{s}$ during the mobile measurements. The measurements were performed on varying routes during daytime to cover the entire city center area.

The CE DOAS is composed of an air sampling system, an optical resonator with two high reflective mirrors, a blue LED light source and a spectrometer (Platt et al., 2009). For the spectral retrieval in the wavelength range $435.6\,\mathrm{nm}$ to $455.1\,\mathrm{nm}$, we used DOASIS (Kraus, 2005). The $NO_2$ reference absorption cross is from Vandaele et al. (2002), $O_4$ from Hermans et al.

(1999), $H_2O$ from Rothman et al. (2003) and CHOCHO (Glyoxal) from Volkamer et al. (2005).

The CAPS measurement technique is closely related to Cavity Ring-Down Laser absorption Spectroscopy (CRDS), which determines the concentration of trace gases from the decay rate of the light source in the optical resonator (Ball and Jones, 2003; Brown et al., 2002; Berden et al., 2000; Engeln et al., 1996). CRDS is a laser-based system, while CAPS uses an incoherent light source (a blue LED) that is well-matched to the $NO_2$ absorption band. The CAPS $NO_2$ system mainly consists of a blue

LED, a measurement chamber with two highly reflective mirrors centered at $450\,\mathrm{nm}$, and a vacuum photodiode detector. It estimates the $NO_2$ concentration by directly measuring the optical absorption of $NO_2$ at the $450\,\mathrm{nm}$ wavelength within the electromagnetic spectrum. The light appears as a distorted waveform after passing through two mirrors and the measurement cell, which is characterized by a phase shift that is determined by demodulation techniques in comparison to the initial LED light modulation. The phase shift is proportional to the absorbance of the light by the presence of $NO_2$. The concentration of

$NO_2$ can be derived by measuring the amount of the phase shift. The detailed principles of the CAPS system are demonstrated in Kebabian et al. (2008, 2005b).

## 2.2  Long-path (LP) DOAS observations

Three LP DOAS instruments were installed on the roof of the MIM. The measurement setups are displayed in Figure 1. The measurement system started operation in December 2015 with a total absorption path of $3828\,\mathrm{m}$ across the English Garden to

a retro reflector array located on the rooftop of the Hilton hotel building at $\sim48\,\mathrm{m}$ height. In January 2017 another absorption path of $1142\,\mathrm{m}$ was installed covering three blocks around the university area to a retro reflector at the St. Ludwig Munich Church at $\sim40\,\mathrm{m}$ above ground. Since July 2015 a retro reflector is also installed at the roof of the N5 building of the Technical University of Munich (TUM) at $\sim28\,\mathrm{m}$ height, allowing an absorption path of $828\,\mathrm{m}$. From July 2016 to August 2017 a path of



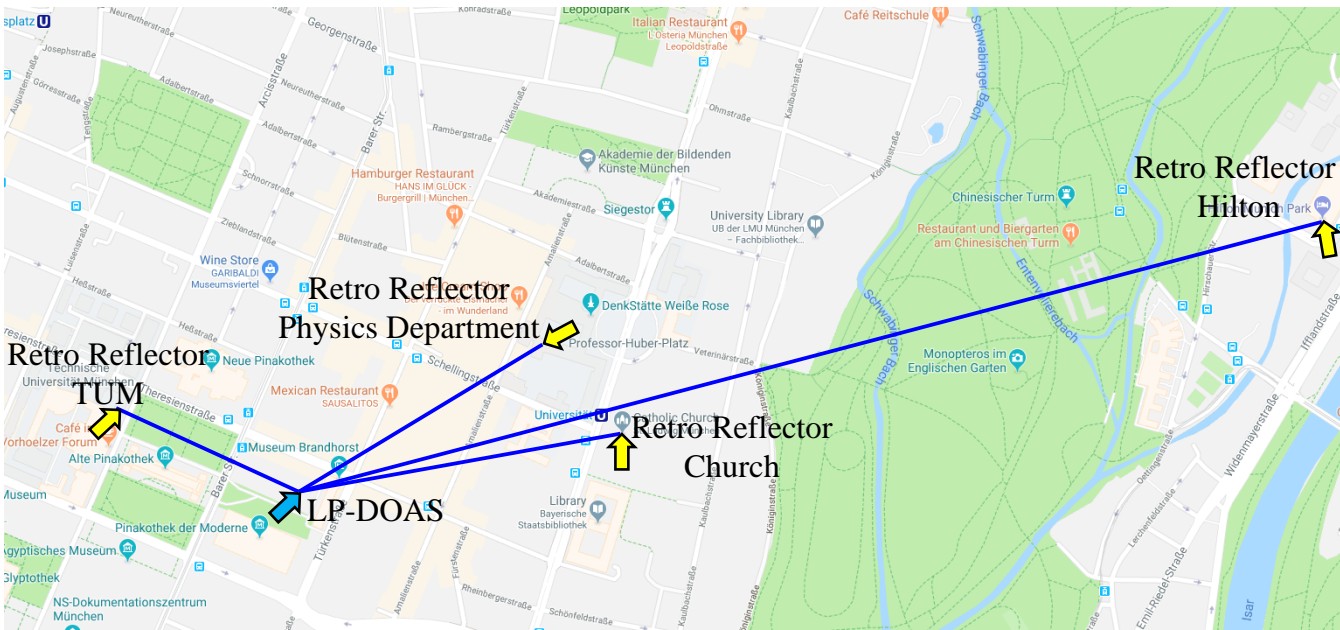

**Figure 1.** Map of Munich city center and four optical paths of three LP DOAS instruments. Map data ©Google maps.

816 m to the roof of the building of the Physics department of LMU at ∼24 m height was operational as well. The measurement paths cover the university campus, the public park, residential areas and areas with heavy traffic. The instrumental background was corrected by subtracting the LED reference spectra, including dark current, offset, and background, from each measured spectrum.

5    A measurement sequence starts by taking a LED reference spectra using a shortcut system consisting of a diffuser plate in front of the y-fiber and an exposure time of 10 s. Then a shutter is used to block the LED for measuring the atmospheric background spectrum for 1 s. Afterwards, the atmospheric spectrum with a maximum of 10 scans is taken. Each scan of a spectrum has a peak intensity of about 60 % to 80 % saturation of the detector and typically requires 60 ms to 1000 ms, depending on the visibility and instrument setup. The total sampling time (the product of the number of scans and exposure

10  time for each scan) was limited to 60 s. A full measurement sequence took between 30 s and 90 s, depending on visibility conditions.

## 2.3   Local air quality monitoring network

The Bavarian LfU is operating five monitoring stations, three roadside stations at Landshuter Allee, Stachus and Lothstrasse, and two ambient stations in Allach and Johanneskirchen. In these stations the air pollutants NO, $NO_2$, CO, $O_3$, $PM_{2.5}$, $PM_{10}$

15  and in addition meteorological parameters such as relative humidity and temperature are measured. In this study we con-



centrated on the NO and $NO_2$ concentrations that are continuously monitored using an in situ CL $NO_x$ analyzer (HORIBA APNA-370) (LfU, 2019).

## 2.4 The Ozone Monitoring Instrument (OMI) satellite observations

The Ozone Monitoring Instrument (OMI) is an imaging spectrometer on board the NASA Aura satellite. It measures earthshine
radiances with two grating spectrometers which cover the wavelength range from $264\,\mathrm{nm}$ to $504\,\mathrm{nm}$. OMI provides the daily measurements of $NO_2$, BrO, $SO_2$, $O_3$, HCHO, OClO, BrO and aerosols in a global coverage. It is able to detect the cloud radiance fraction, cloud pressure and albedo.

NASA's OMI standard product version 3 (SPv3) (Krotkov et al., 2017; Marchenko et al., 2015) is used in this study. The OMI $NO_2$ data is publicly available at the Goddard Earth Sciences Data and Information Services Center (GES DISC)
(https://disc.gsfc.nasa.gov/). For our comparison we gridded the OMI VCDs onto a high resolution grid with $0.02° \times 0.02°$ following the approach described in previous studies (Chan et al., 2015; Kuhlmann et al., 2014).

### 2.4.1 Converting OMI vertical column densities to ground mixing ratio using modeled $NO_2$ profiles

In order to examine how representative the LP DOAS data is for the temporal pattern observed by OMI, which covers the entire city, OMI's vertical column densities (VCDs) are converted into ground concentrations. For the conversion, vertical
profile information is needed. We utilized $NO_2$ vertical profile information simulated by the chemistry transport model (CTM) GEOS-Chem (Bey et al., 2001). The horizontal resolution of the simulation is $2.0°$ (latitude) $\times 2.5°$. Vertical profiles of $NO_2$ are spatially interpolated within the 4 closest grid cell to the measurement location. Detailed description of the GEOS-Chem simulation can be found in previous studies (Chan, 2017a, b).

Since we use the LP DOAS data for the diurnal correction, we tested the correlation between LP DOAS measurement values
and OMI data. The OMI satellite measurements cover a larger area of Munich with the instrument's ground pixel footprint of $\sim 320\,\text{-}\,6400\,\mathrm{km}^2$ than the $2\,\mathrm{km}$ path length of the LP DOAS instrument. A good correlation would allow the assumption that the relative diurnal cycle obtained from the LP DOAS can be regarded as being representative for the entire urban area.

## 3 Results and Discussion

### 3.1 $NO_2$ concentration maps constructed using mobile measurements

The mobile measurement data can be used to create a map showing the city-wide distribution of air quality using $NO_2$ concentrations as a general indicator (Figure 3). As a first test, we compared the averaged measurement values within a $10\,\mathrm{km}$ radius around the three governmental monitoring stations at Landshuter Allee, Lothstrasse and Stachus and obtained an averaged concentration of $93\,\mathrm{\mu g/m^3}$ for the mobile measurements and $48\,\mathrm{\mu g/m^3}$ for the three stations for the campaign days in June and July 2016. The large difference can be explained by looking at the criteria for the location of monitoring sites set by the
European Union: the recommended measurement height is between $1.5\,\mathrm{m}$ and $4\,\mathrm{m}$, maximum distance to the street is $10\,\mathrm{m}$

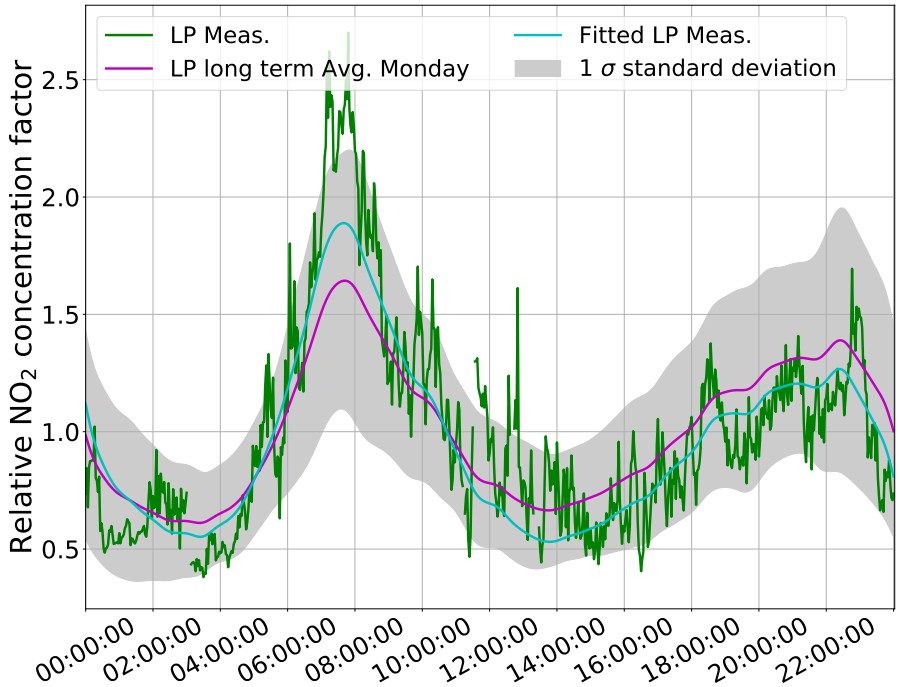

**Figure 2.** Normalization curve used to correct mobile measurement data. The purple curve is the long term average diurnal pattern for the day of the week of the measurement day (Monday in this example), which is fitted (scaling with linear time dependent factor and offset) to the measurement data of $13^{th}$ June 2016 shown in green, excluding the data outside of the $2\,\sigma$ area shown in gray. The resulting cyan curve is used to remove the diurnal dependency of the mobile measurements data.

and at a minimum distance to the next crossroad of $25\,\mathrm{m}$ (Commission, 2008). Most monitoring stations have the inlet positioned at $4\,\mathrm{m}$ height. The mobile measurement data, however, include the significantly increased concentrations at crossroads, tunnel exits and other pollution hot spots. In addition, the height of the measurement inlets differs by $2.5\,\mathrm{m}$ between the mobile measurements and the governmental monitoring stations, which also influences the comparison. In order to determine how representative point measurements are for the city-wide air quality, we analyzed the correlation between point measurements and the distribution captured by mobile measurements, between point and path averaging measurements, and between path averaging and satellite measurements. Since the spatial distribution can not be captured instantaneous, an algorithm to normalize for the diurnal variation is needed in order to create a consistent map representing only the spatial variability of daily average concentrations instead of temporal influences.



### 3.1.1 Normalization of the diurnal cycle

As a mobile survey cannot capture the concentrations at different locations simultaneously, and the $NO_2$ measurements are naturally influenced by daily variations such as changing boundary layer height or the diurnal cycle of the traffic amount, we use an algorithm to separate temporal and spatial patterns in the data set.

First, the algorithm normalizes the long time series of LP DOAS measurements of atmospheric $NO_2$ by dividing through the daily average $NO_2$ concentration of the same day. The mean concentration curves for each day of the week over a period of 2.5 years are calculated in order to obtain a relative diurnal $NO_2$ variation pattern (purple curve in Figure 2). The normalized averaged diurnal $NO_2$ curve of the corresponding weekday is fitted (using an offset and a scaling with a linearly time dependent factor) to the normalized LP DOAS measurement of the corresponding day, coinciding with the mobile measurements. In order

to remove the influence of outliers, $NO_2$ values outside of $2\sigma$ variation of the fitted curve are disregarded (cyan curve). Figure 2 shows the fitting process for the normalization curve for one day of the measurement campaign. The other days show very similar characteristics with a significant peak in the morning and evening rush hours. Dividing the mobile measurement data by the curve data, removes the diurnal dependencies and allows focusing on spatial pattern.

### 3.1.2 Spatial distribution of $NO_2$ in the city of Munich

The measured concentrations during the campaign were spatially averaged to a high resolution grid of $20\,\text{m} \times 20\,\text{m}$ (Figure 3a). Most of these measurements are distributed on major roads, including city, urban ring-road, suburbs, rural areas, and highways. Relatively high $NO_2$ pollution could be observed on motorways and busy urban roads. Difference between main roads and adjoining side roads of up to a factor of 5 can be observed. $4.4\,\%$ of the on-road measurements exceeded the WHO 1-hour guideline value of $200\,\mu\text{g/m}^3 \approx 106\,\text{ppb}$ (depending on temperature, here the appropriate conversion factor at $25\,°\text{C}$

and $1013\,\text{hPa}$ are used), corresponding to $6.6\,\%$ of the area covered. High $NO_2$ values over motorways were mainly due to the emission of heavy duty diesel vehicles, i.e. a significant increase could be observed when we were driving behind trucks and buses. In the city center traffic congestion and the street canyon effect (Rakowska et al., 2014) can be the main cause of elevated on-road $NO_2$ levels. Zhu et al. (2018) showed in a study in Hong Kong that average pollution exposure increases by $14.5\,\%$ when stopping at a traffic light compared to fluent traffic. Other studies showed as well that the distribution of

pollutants is mainly impacted by traffic flow patterns (Fu et al., 2017; Rakowska et al., 2014; Huan and Kebin, 2012; Kaur et al., 2007; Westerdahl et al., 2005). The normalization using coinciding LP DOAS measurement removes the diurnal dependency but leaves the traffic flow dependency in the data, because it contributes to the city-wide air quality. The normalized on-road $NO_2$ map shown in Figure 3(b) represents daily average values for all locations. After normalization, there exist some regions, where $NO_2$ concentrations are consistently higher, while in other areas the normalized concentrations are lower than the original measurements. This behavior can be explained by the time of the day when the measurements were taken: when

we measured during the rush hour, the measurements are higher, while the measurements during noon are lower than the daily average. The normalization procedure increased the occurrences of WHO 1-hour guideline exceedances to $14.5\,\%$ of the on-road measurements, corresponding to $17.1\,\%$ of the total area (including motorways) and $15.7\,\%$ of the area in the city center.





**Figure 3.** (a) CE DOAS and CAPS mobile measurements of NO$_2$ in Munich in 2016. (b) Normalized spatial distribution of NO$_2$ using coinciding LP DOAS data to remove the diurnal dependencies. (c) Zoom in of the city center. The three black diamonds in (c) are the locations of the governmental monitoring stations (Landshuter Allee, Lothstraße, Stachus). The area at the top right with very low concentrations represents the city park English Garden. Map data ©Google maps.

However, the thresholds in WHO AQG are based on studies involving monitoring station data which are not measuring directly on the street. Taking the vertical and horizontal dilution factors (see Figure 4, the factor 0.84 for 4 m height and 0.87 for 6 m distance) into account, we extrapolate WHO AQG 1-hour threshold value of $200\,\mu g/m^3$ to the on-road level with the value of $273.7\,\mu g/m^3 \approx 145.6\,ppb$ ($200\,\mu g/m^3/0.84/0.87$)), then also calculated the frequency of exceedances (cf. table 1).



**Table 1.** Percentage of measured concentrations exceeding the WHO AQG and its corrected on-road level thresholds for both temporal and spatial coverage. The values are broken down for before and after the normalization of the data according to diurnal patterns, and also calculated for WHO guideline values adjusted for the different measurement height and distance to the street.

|  | Percentage exceeding WHO guideline | | Percentage exceeding adjusted WHO guideline | |
| --- | --- | --- | --- | --- |
|  | Before normalization | After normalization | Before normalization | After normalization |
| Temporally | 4.4 % | 14.5 % | 1.7 % | 6.7 % |
| Spatially (total) | 4.1 % | 17.1 % | 1.1 % | 4.6 % |
| Spatially (downtown) | 5.5 % | 12.4 % | 1.6 % | 4.7 % |

It can be seen that especially in the downtown area (Figure 3c), the values after the normalization are noticeable higher than before. This can be explained as we tried to avoid the rush hours, i.e. traffic jams, for performing the measurements. Therefore, the measured $NO_2$ level is often lower in comparison to the day-average. The area with significantly lower concentrations seen in Figure 3 is the city park (English Garden) with no vehicle emissions and where plants could provide deposition areas for $O_3$

, $NO_x$ and particles (Chaparro-Suarez et al., 2011; Wesely and Hicks, 2000).

### 3.2 Comparison of $NO_2$ concentrations at different heights and distances from the street

In order to investigate the diffusion effects of emitted $NO_2$ molecules in both vertical and horizontal directions, measurements were conducted over two days (20th. and 22th. March 2019) using CE DOAS and CL instruments at Stachus, Munich, next to the governmental monitoring station. Since we used two different measurement techniques, the first step was to check the

instrument for consistency. Side-by-side measurements next to the street (same height and distance to the street in Figure 4) were used to analyze differences. We found the CL $NO_2$ to be 2 % higher, possible due to sensitivities of the molybdenum oxide converters to $NO_y$ species (see Villena et al., 2012; Dunlea et al., 2007). We corrected the CL measurement data in order to remove those interferences.

Both instruments were set up next to the governmental monitoring station at Stachus, which is at a height of 4 m and has

a 30 min time resolution. The CE DOAS was set up next to the street and measured at a fixed height of 1.5 m above the ground, while the CL instrument measured $NO_2$ at multiple heights above the ground (from 0.5 m to 4 m) and at different distances from the side of the street (from 2 m to 10 m). The temporal resolution for both instruments was set to 5 s. All measurements are shown in Figure 4(a) for the different measurement heights and (b) for the different distances to the side of the street. Figure 4(c) shows the distribution of the ratios, and it can clearly be seen that the average concentrations decrease

with height and distance. Figure 4(d) shows a two-day comparison between the 30 min average $NO_2$ concentrations measured with our CE DOAS at a height of 1.5 m with the CL instrument data of the governmental monitoring station at a height of 4 m. The regression plot shows a ratio of 1.23 between the measurements at 1.5 m and 4 m height. We repeated the 1.5 m to 4 m measurement height comparison on several different days at different seasons and derived the same factor of 0.84 with





a standard deviation of 0.21, so most ratios vary from $37\%$ decrease with increasing height (factor 0.84-0.21=0.63) to $5\%$ increase (factor 0.84+0.21)=1.05). Since the inlet height for our mobile measurements is $1.5\,\mathrm{m}$, we take this factor into account when comparing to monitoring station data. In terms of distance to the street, measuring at the center of the street, like we did during the mobile measurements, and measuring at a distance of $6\,\mathrm{m}$, which is approximately the distance of most monitoring

stations to the middle of the street, the on-road measurements are $13\%$ higher due to the observed diffusion effects. Those factors have to be kept in mind when comparing on-road measurements to monitoring station data or any other measurement data taken at different height levels and distances to the street. This leads to the conclusion, that from the $48\%$ difference between the average concentrations of three monitoring stations ($48\,\mu\mathrm{g/m}^3$) and the mobile measurements around the three stations ($93\,\mu\mathrm{g/m}^3$), both averaged for the measurement campaign period, $27\%$ can be explained by the difference in inlet

height and distance to the street, and the remaining $21\%$ is due to the fact, that the monitoring stations are positioned away from pollution hot spots at crossroads according to WHO guidelines.

### 3.3   Comparison between ground measurement and converted OMI observation

Satellite measurements are strongly affected by clouds, as clouds shield ground level $NO_2$. Hence, to compare with the LP DOAS data, OMI data with cloud fractions larger than $50\%$, which were significantly influenced by clouds, were filtered out.

LP DOAS data from 12:00 - 15:00 UTC, same as the OMI overpass time for Munich, were used for comparing with the average OMI data sets within $10\,\mathrm{km}$ and $50\,\mathrm{km}$ from the measurement site (Figure 5). To reduce the impact of clouds and local spatial variations, we use monthly average data to compare. The uncertainty of the LP DOAS measurements, which was smaller than $1\,\mathrm{ppb}$ for a single measurement, is too small to be shown for monthly averages so the standard deviation within each month is shown in Figure 5. Observation of LP DOAS and OMI both showed a similar annual trends, with higher $NO_2$ levels in winter

and lower $NO_2$ levels in summer.

Comparing monthly means of LP DOAS overpass time measurements with OMI retrieved ground mixing ratios of $NO_2$ and VCDs within $10\,\mathrm{km}$ show correlation coefficients of 0.85 and 0.72, respectively (Figure 6). OMI and the corresponding overpass time measurements of LP DOAS correlate well, indicating OMI measured reliable tendency of ground level $NO_2$. The discrepancy of correlation coefficients were mainly caused by the vertical profile used for the OMI retrieval and the

conversion of VCDs to ground level mixing ratios. In order to show the influence of temporal averaging and assess the temporal representativeness of the governmental monitoring station data, monthly averages for all LP DOAS data and the data of three governmental monitoring stations are shown in Figure 5 as well. In addition, the data of the governmental monitoring stations were $15.4\,\mathrm{ppb}$ on average higher than the LP DOAS measurements. The Pearson correlation between the two data sets was 0.32. The low correlation may be due to the different measurement areas, measurement heights and measurement resolution.

Average variability of the $NO_2$ mixing ratios in winter (November to February) and summer (June to August) were $16.5\,\mathrm{ppb}$ and $5.5\,\mathrm{ppb}$. The errors bars of OMI data do not overlap with the LP DOAS data for most months, neither within $10\,\mathrm{km}$ nor within $50\,\mathrm{km}$, which demonstrated that there might be systematic errors in the conversion of VCDs to ground level mixing ratios or in the OMI retrieval process itself (Wenig et al., 2008). As the previous study also suggests, lower OMI values over cities compared to ground measurements could be due to the OMI a-priori profile used for both, the VCD retrieval and

**Figure 4.** NO$_2$ measurements at Karlsplatz (Stachus), Munich, with different heights above ground (a) and distances to the main street (b). The ratio between 1-minute averaged CE DOAS and coinciding CL measurements were calculated individually for the different heights and distances. Half-hour averaged CE DOAS measurements (blue curve) were compared with the corresponding CL measurements (red curve). Averaged ratios for different heights and distances (as defined in (a)) are shown in (c), CE DOAS measurements at $1.5\,\mathrm{m}$ were averaged to $30\,\mathrm{min}$ intervals and compared with the half-hourly data of the governmental monitoring station at $4\,\mathrm{m}$ shown in (d)

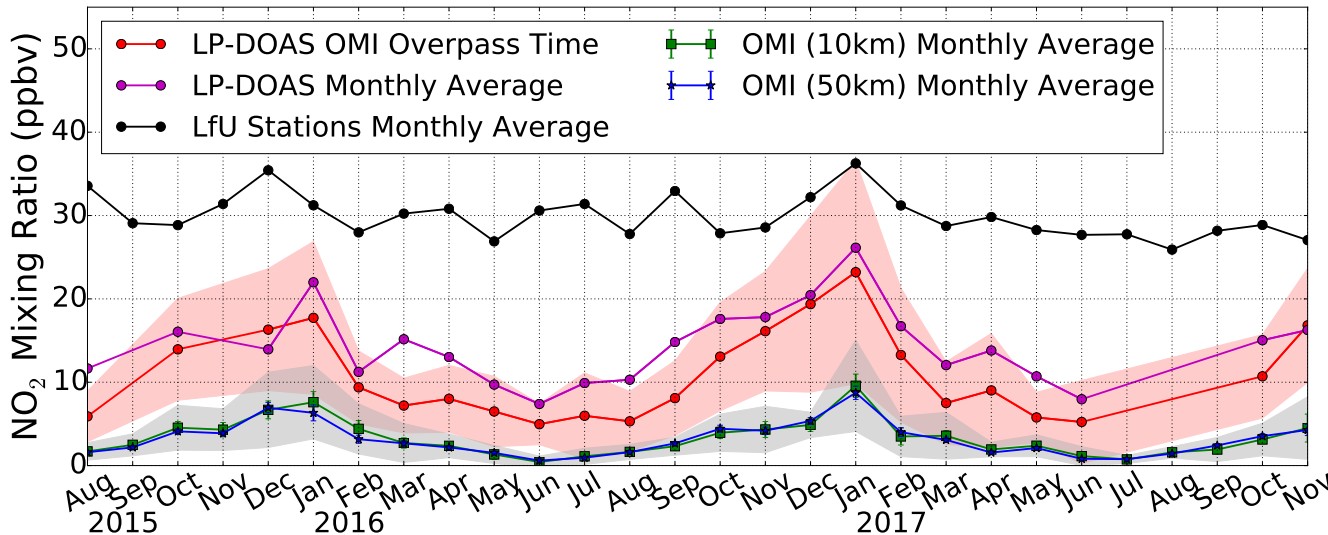

**Figure 5.** Monthly average ground-level NO$_2$ mixing ratio. The monthly average NO$_2$ of the governmental monitoring stations (Landshuter Allee, Lothstraße, Stachus) is shown. LP DOAS data are temporal averages around OMI overpass time (12:30 – 14:30 UTC). OMI data are spatial averages over pixels within a 10 km and 50 km radius of our institute. The red shadow indicate the variability of 1 $\sigma$ (standard deviation) of averaged LP DOAS measurements of OMI overpass time. The light gray regions indicate 1 $\sigma$ variability of OMI data within 10 km

converting the tropospheric NO$_2$ VCDs to ground level mixing ratios, was taking an average over a larger area, not only urban areas, but also rural areas with a lower ground level mixing ratio to total column ratios. A total underestimation for the ground level NO$_2$ of about 69 % can be observed. A similar result was found by Kuhlmann et al. (2015). However, because of the good correlation it is safe to assume that relative temporal changes captured by the LP DOAS can be regarded as representative for area covered by OMI that spans the entire city of Munich.

## 4 Summary and conclusions

Mobile road measurements using CE DOAS and CAPS instruments combined with an algorithm for correcting the diurnal cycle were used in order to generate a consistent pollution map of the street level NO$_2$ concentration in Munich. This map is not only used to identify pollution hot spots but also to figure out how representative the existing NO$_2$ point measurements are for the whole city. Elevated NO$_2$ levels can be observed mostly on motorways and busy city roads, due to the emission of heavy duty vehicles or heavy traffic volume. When averaging the mobile measurements around the monitoring stations, we derived an average NO$_2$ concentration of 93 μg/m$^3$, whereas the three monitoring stations at the city center reported 48 μg/m$^3$ on average for the same time, so 48 % lower values. Our analysis shows that the different measurement height can account for 16 % difference (factor 0.84), and the distance of the sample inlets to the center of the street, where the mobile measurements took



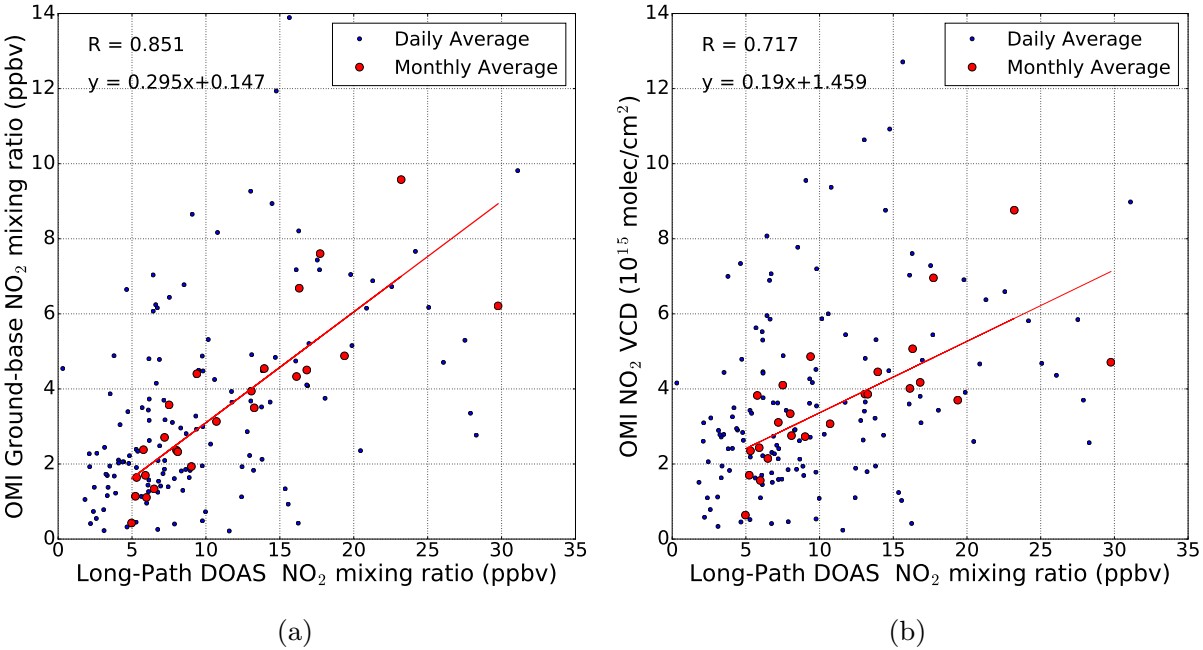

**Figure 6.** Correlation of LP DOAS measurements with retrieved NO$_2$ mixing ratio (ppbv) in (a) and VCD in (b) of OMI observation.

place explains the 13 % (factor 0.87) lower values. Accounting for these factors still leaves about 21 % that can be attributed to pollution hot spots like busy cross roads or tunnel exits. These hot spots are not covered by monitoring stations which is intentionally done in order to make the long term data less dependent on local events. Nevertheless, the differences observed in the presented study shows that point measurements are likely not representative for the NO$_2$ concentration in the whole city.

5   Most network measurement sites are not capturing the concentrations people are exposed to when walking or driving at street level but are instead focusing on long term trends. Our study illustrates the importance of combining different measurement techniques to capture spatial and temporal patterns within a city and derive concentration values that are representative for the air most people breathe in.

The pollution maps generated in this project provide valuable information for future urban planning and the design of

10  control measures of environment pollution. Furthermore, it can provide guidelines for identifying representative locations for air pollution monitoring stations in a city. Additionally, the observed spatial distribution of NO$_2$ concentrations are also beneficial to the validation of chemical transport models and assessment studies of the impact of air pollution on human health.

*Author contributions.*   YZ, MW and JC designed the experiments. YZ, GK and XB carried them out. KLC simulate the results from GEOS-Chem model. YZ prepared the manuscript with contributions from all co-authors.



*Competing interests.*   The authors declare that they have no conflict of interest.

*Acknowledgements.*   The work described in this paper was jountly supported by the major research instrumentation programme INST 86/1499 FUGG.





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
