# Peer review of "Spatial and temporal representativeness of point measurements for nitrogen dioxide pollution levels in cities"

_Atmospheric Chemistry and Physics, 2019_

## Referee Comment (RC1) · Anonymous Referee #1 · 15 Apr 2020

This manuscript presents an approach to merge a combination of mobile and stationary in-situ observations of NO2 with ground-based remote sensing of NO2 to produce a temporally consistent NO2 concentration surface within an urban area at 20m x 20m spatial resolution. Using this surface, they infer the frequency of exceedances during the sampling campaign relative to current WHO NO2 guideline concentrations. The authors discuss mismatches between the sampling location at typical monitoring stations and concentrations observed on roads and at street level, and conclude that a variety of monitoring techniques are important to derive representative air pollutant concentrations.

[Figure]

The authors can be commended for their attempt to to derive an approach that synthesizes NO2 observations across multiple relevant scales, from the micro environment to integrated urban scales. They discuss several important insights that may be of interest to other investigators who measure NO2 with in-situ and remote sensing techniques. There are a number of novel measurements described here, and the authors have put in a great deal of effort to interpret them clearly. However, several elements of their methods and analysis could benefit from further clarification and evaluation, as I outline below. I have a number of general concerns that should be addressed before publication, followed by a list of specific comments.

General comments:

(1) Mobile measurements: More details would be very helpful. It's not clear at all how many, when, and what routes were taken for the mobile measurements. The authors simply state that mobile measurements "were performed in June and July 2016" and that "varying routes" were taken. Were mobile deployments performed every day over those two months? What were the routes like? I suggest the authors at least provide a table of the dates and times that mobile measurements were made, and a map of each of the routes (unless they were all identical, or the number of mobile deployments was truly large) as Supplementary Information, and include a short description in the main text (at least N=?). Since I assume the route was not identical in each case, it might be useful to include a panel in Figure 3 showing how many spatially overlapping measurements are included in each pixel average (i.e. how many days are included in each 20 x 20m average?). Also, it's not clear until much later in the manuscript that the authors explain these observations are gridded to 20 x 20 m. This should be included in the description here.

(2) Long path DOAS: I'm also a little confused about how all the different LP-DOAS measurements were used. The methods describe four different absorption paths, shown in Figure 1. But when/how is each absorbing path used? It seems to me that the LP-DOAS measurements are only used in the normalization step (Section 3.1.1).

[Figure]

Were all the LP-DOAS measurements averaged together to get each day's diurnal cycle? If so, how? Or were different absorption paths used for different parts of the city? Or were the results from only one absorption path used? (If the latter, why are all paths described and shown?)

(3) Diurnal normalization: In my reading of this manuscript, the algorithm for diurnal "normalization" based on LP-DOAS measurements above the city street levels strikes me as one of the central developments presented in this manuscript, underpinning their main results. But the approach is not evaluated in any way. In my opinion, some evaluation is warranted, since it is a key aspect of their final results. I think a satisfactory evaluation could be quite easy to accomplish using the available ground-based local air quality monitoring data for which you have full diurnal observations. For example, I would propose: Why not take a random sampling of NO2 measurements at various dates and times from the stationary monitoring sites, and see if the correct diurnal profile/24-hr average can be retrieved based on the LP-DOAS normalization for that date. Repeating this experiment for, say, 50 different days from different times of day, at all three of the monitoring stations at coincident times, could quite nicely result in statistics describing how accurate this diurnal normalization approach is, depending on the time and location sampled (at least for these stationary cases). Also, just to be clear, the output of this algorithm is essentially a "retrieved" 24-hour average concentration from a measurement at any instant in time, correct? I think this is worth spelling out very clearly. In other words, instead of calling it a "normalized" concentration, why not call it an "inferred daily average" or something like this? This would be much more intuitive. Perhaps these are not mathematically the same? (But if not, why can these numbers be compared to the WHO 24-hour average guideline concentration?)

(4) Comparison of NO2 concentrations at different heights and distances: I may not have understood correctly, but is this section based on measurements from a single monitoring station over two individual days? At the beginning of Section 3.2, the authors state "measurements were conducted over two days" in March, but then later in

the same section, the authors state "we repeated the 1.5 m to 4 m measurement height comparison on several days at different seasons and derived the same factor". This is confusing and requires clarification. Please be specific regarding the deployments. As written, it seems like the authors have derived the rules for vertical and horizontal $NO_2$ diffusion based on one specific monitoring location from two days in March, and applied that generally for their mobile results across the whole city taken in June and July. Wouldn't the dilution and chemistry of $NO_2$ be specific to each micro environment, where the monitoring is done? Buildings, wind patterns, light availability, I would expect to all play a role in the spatial scale of $NO_2$ concentrations at such small scales, which would vary depending on location. Likewise, I'm surprised that these spatial scales are independent of local meteorology, and seasonality (driving $O_3$ concentrations and other factors for example). I think this requires some evaluation, clarification, and some reasonable caveats. Finally, isn't there prior work to cite, related to how $NO_2$ is observed to decay by distance from, say, a highway? How do your results compare? Are the spatial scales that different?

(5) Comparison with OMI: It's not clear to me what is gained from the comparison with OMI in Section 3.3, and how it is relevant to this study. Essentially, the authors have demonstrated that the LP-DOAS and OMI observations share seasonality. They also conclude that the OMI-inferred values are lower than the LP-DOAS values. I don't see much new here, but perhaps the authors could clarify. How exactly does this comparison support the main conclusions of this manuscript, which involve extremely small spatial scales within a small period of time that would not be well captured by OMI?

(6) Main result: In my opinion, one of the main results of this manuscript is how many exceedances of the WHO guideline would actually be inferred with a more dense monitoring strategy, compared to the sparse observations currently available. But this result gets glossed over in the rest of the discussion and concluding remarks. I'd be curious if the authors agree that this one of the most important results, and if they think it

deserves more prominent position in the abstract/conclusion.

Specific comments:

Introduction:

Page 2, Line 11: "NO2 levels are often strongly correlated with many other toxic air pollutants". Perhaps this is indeed well-known, but I think this statement merits some citations.

Page 2, Line 25: "measures" should be "measure"

Page 2, Line 25: The authors refer to "defined rules" in terms of measurement height and distance in several places throughout the manuscript, but have mentioned what these rules are anywhere. I assume these "rules" may vary by jurisdiction. Are there consistent, e.g. European-wide, rules? Does the WHO provide location guidelines? I think at least providing the "rule" for the local agency would be insightful to share somewhere. Even better, share the rules for other places if they vary.

Page 3, Lines 15-20: It seems to me that certain portions of the introduction, including these lines, would be better placed in the Methods section of this manuscript

Methods:

Page 5, Line 13: The names of each urban station means very little to someone unfamiliar with the region. I later discovered these locations are indeed on the map in Figure 3c. I suggest the authors mention here that these locations can be found on this map.

Page 6, Lines 4-11: The authors explain that the OMI VCDs are gridded onto a 0.02 x 0.02 grid following approaches from previous work, and later in the manuscript it looks like the authors use monthly averages. Can the authors be slightly more clear here? Are the observations from each day gridded separately, then I assume simple monthly averages are calculated from these daily surfaces?

Page 6, Lines 13-20: Can the authors be clear about what model profiles are being used? I.e., what time of day, and I assume from the same month/year as the satellite observations? Is the surface estimate calculated each day, prior to calculating the monthly average? Or is the surface estimate performed on the monthly average columns? I believe the authors answered these questions in the response to the quick review, but why not include these details in the text for the benefit of readers? Regarding the conversion to surface concentrations, can the authors describe the logic behind using spatially interpolated profiles across four grid cells? Doesn't this represent interpolating over an area of 4x5 degrees? I am curious why they would not simply use the spatially coincident model grid cell.

Page 6, Lines 21-22: I'm not sure I agree entirely here. What if the diurnal pattern of sources or chemistry is different over the area of the satellite pixel, compared to the area covered by the LP-DOAS absorption path. Wouldn't that cause a mismatch between the diurnal cycle integrated over each?

Results and Discussion:

Page 7, Line 11: "as a general indicator". An indicator of what?

Page 7, Line 13: Why have the concentrations been averaged over the three locations in this case, which presumably represent quite different micro environments? Why not just report the average concentration of each separately, and within each of their 10-km radius? Section 3.1.1: As I mention above, this really strikes me as belonging in the Methods section. This means you could also keep Figure 2 before Figure 3. As it is now, you reference Figure 3 in the text before discussing Figure 2.

Figure 3c: You could be very clear here that this is a zoomed-in plot of the diurnally normalized data.

Table 1: You could be very clear here that the WHO AQG is according to the 24-hr average guideline.

Page 10, Line 17: This is assuming that the WHO guidelines are based on the same monitoring "rules" that are applied by the environmental reporting agency. Is this true? What are the WHO guidelines for monitor placement? Are these the same rules employed by the local air quality agency? Please share these rules explicitly for the benefit of the readers.

---

## Referee Comment (RC2) · Anonymous Referee #3 · 18 May 2020

This paper has two themes. First it analyses spatial variation in urban NO2 based on mobile and stationary point measurements. Then it describes remote sensing observations. I recommend a major revision prior to publication.

Taking each in turn.

1) The mobile data set is interesting. However other mobile data sets examining spatial variability in cities exist and it is not clear if any of the observations here are surprising because they are not placed in the context of the prior related literature. I recommend adding a deeper and more comprehensive discussion of the theoretical and observational understanding we have of emissions from roadways and the length scales

of decay of those emissions in cities. For example, papers by Choi et al. including https://doi.org/10.1016/j.atmosenv.2012.07.084 and Atmos. Chem. Phys., 14, 6925–6940, 2014 and by Apte et al. Environ. Sci. Technol. 2017, 51, 12, 6999–7008 show a characteristic decay length scale of 500-1000m (1/e) that would be approximately consistent with the measurements reported herein. The Choi et al ppaers also provide a theoretical basis for discussion of the decay. It is also important to note that the time scale for conversion of NO to NO2 is not instantaneous. Thus on-road measurements of NO2 may have a systematic bias. The measurements in Apte, et al. show the consequences of NO to NO2 conversion as different timescales/lengthscales for decay from urban roadway sources.

2) I find the discussion of the remote sensing measurements confusing. The logic connecting them to the mobile measurements is unclear. It is well-known that OMI measurements with a 2 degree a priori will have a large bias compared to urban measurements. The large context of the long path measurements connects both emissions and loss, while the mobile measurements are so near to the source that they only reflect emissions. I recommend these sections be removed or the connection to the mobile observations made substantially clearer.

Also, to help with the readability of the paper, I recommend moving all descriptions of the instruments to the supplement.

---

## Author Comment (AC1) · 21 Jul 2020

We sincerely thank the reviewer for providing her/his very valuable suggestions, which helped us improve our paper significantly. Below please find our responses:

General comments:

(1) Mobile measurements: More details would be very helpful. It's not clear at all how many, when, and what routes were taken for the mobile measurements. The authors simply state that mobile measurements "were performed in June and July 2016" and that "varying routes" were taken. Were mobile deployments performed every day over
those two months? What were the routes like? I suggest the authors at least provide a table of the dates and times that mobile measurements were made, and a map of each of the routes (unless they were all identical, or the number of mobile deployments was truly large) as Supplementary Information, and include a short description in the main text (at least N=?). Since I assume the route was not identical in each case, might be useful to include a panel in Figure 3 showing how many spatially overlapping measurements are included in each pixel average (i.e. how many days are included in each 20 x 20m average?). Also, it's not clear until much later in the manuscript that the authors explain these observations are gridded to 20 x 20 m. This should be included in the description here.

As the reviewer suggested, detailed description regarding hours per day of all the measurement days and the total amount of measurements per day have been listed in Table 1, Section 2.1.

All measurements were spatially averaged to a high-resolution grid of 20m x 20m. This resolution has been chosen according to an average driving speed of 40-50 km/h and the measurement sampling time of 2s, so that at least one measurement value per drive-by falls into each grid box. All measurements from different drives and different days are averaged for each grid cell after diurnal cycle normalization.

This description was added in the paper (Section 2.1, line 10-15).

(2) Long path DOAS: I'm also a little confused about how all the different LP-DOAS measurements were used. The methods describe four different absorption paths, shown in Figure 1. But when/how is each absorbing path used? It seems to me that the LP-DOAS measurements are only used in the normalization step (Section 3.1.1).3 Were all the LP-DOAS measurements averaged together to get each day's diurnal cycle? If so, how? Or were different absorption paths used for different parts of the city? Or were the results from only one absorption path used? (If the latter, why are all paths described and shown?)

For the long-term average diurnal cycle calculation described in Sec.3.1.1 all available data from all measurement paths were used. LP DOAS measurements of light paths towards to Hilton hotel, TUM University, and LMU physic department were used to normalize the mobile measurements regarding the diurnal cycle. The same description was added in the paper (Section 2.2, line 19-22).

(3) Diurnal normalization: In my reading of this manuscript, the algorithm for diurnal "normalization" based on LP-DOAS measurements above the city street levels strikes me as one of the central developments presented in this manuscript, underpinning their main results. But the approach is not evaluated in any way. In my opinion, some evaluation is warranted, since it is a key aspect of their final results. I think a satisfactory evaluation could be quite easy to accomplish using the available ground-based local air quality monitoring data for which you have full diurnal observations. For example, I would propose: Why not take a random sampling of $NO_2$ measurements at various dates and times from the stationary monitoring sites, and see if the correct diurnal profile/24-hr average can be retrieved based on the LP-DOAS normalization for that date. Repeating this experiment for, say, 50 different days from different times of day, at all three of the monitoring stations at coincident times, could quite nicely result in statistics describing how accurate this diurnal normalization approach is, depending on the time and location sampled (at least for these stationary cases). Also, just to be clear, the output of this algorithm is essentially a "retrieved" 24-hour average concentration from a measurement at any instant in time, correct? I think this is worth spelling out very clearly. In other words, instead of calling it a "normalized" concentration, why not call it an "inferred daily average" or something like this? This would be much more intuitive. Perhaps these are not mathematically the same? (But if not, why can these numbers be compared to the WHO 24-hour average guideline concentration?)

The reviewer has correctly understood our normalization algorithm. Thanks to his/her suggestion, we replaced the description of "normalized" into "inferred daily average" when possible.

The corresponding stationary measurements from three LfU stations on mobile measurement days were used to evaluate the accuracy of the normalization algorithm. The inferred daily average concentration for each stationary measurement was retrieved based on the fitted LP DOAS normalization curve, then compared with the corresponding 24-hour average concentration measured by the same station. The distribution of the concentration differences is shown in Fig. 1 below. The averaged difference is 0.7ppb with a 1 sigma uncertainty of 5.21ppb. The inferred daily averages compared to the actual average concentrations at the stations Landshuter Allee and Stachus differ more because due to frequent traffic jams and stop-and-go traffic they show a different diurnal cycle. Overall, the normalization algorithm effectively separated the temporal and spatial influence for most mobile measurement locations. We added the same description in the paper (Section 3.1.1, line 15-20).

(4) Comparison of NO2 concentrations at different heights and distances: I may not have understood correctly, but is this section based on measurements from a single monitoring station over two individual days? At the beginning of Section 3.2, the authors state "measurements were conducted over two days" in March, but then later in the same section, the authors state "we repeated the 1.5 m to 4 m measurement height comparison on several days at different seasons and derived the same factor". This is confusing and requires clarification. Please be specific regarding the deployments. As written, it seems like the authors have derived the rules for vertical and horizontal NO2 diffusion based on one specific monitoring location from two days in March, and applied that generally for their mobile results across the whole city taken in June and July. Wouldn't the dilution and chemistry of NO2 be specific to each micro environment, where the monitoring is done? Buildings, wind patterns, light availability, I would expect to all play a role in the spatial scale of NO2 concentrations at such small scales, which would vary depending on location. Likewise, I'm surprised that these spatial scales are independent of local meteorology, and seasonality (driving O3 concentrations and other factors for example). I think this requires some evaluation, clarification, and some reasonable caveats. Finally, isn't there prior work to cite, related to how NO2

is observed to decay by distance from, say, a highway? How do your results compare? Are the spatial scales that different?

The full analysis for NO2 concentrations at different heights and distances has been done for 2 complete days at one station as described in the paper. Additional roadside measurements regarding the comparison between the 1.5m and 4m measurement height were conducted next to LMU Building and Landshuter Allee on several different days during different seasons and derived a similar result. Besides, the reviewer has a good point, those scaling factors depend on a lot of other factors, local meteorology, and seasonality as he/she suggested, but also local topography, driving speed and speed of the other cars, and so on. We analyzed how the uncertainty of the scaling factors contribute to the comparison with the WHO threshold shown in Table 2 and found that the percentage of exceedances can vary between 0.4% and 23.8%. We don't assume that the scaling factors can be applied to all our mobile measurements but rather on average for our statistical analysis. Modified description locates in Section 3.2, line 1-5.

(5) Comparison with OMI: It's not clear to me what is gained from the comparison with OMI in Section 3.3, and how it is relevant to this study. Essentially, the authors have demonstrated that the LP-DOAS and OMI observations share seasonality. They also conclude that the OMI-inferred values are lower than the LP-DOAS values. I don't see much new here, but perhaps the authors could clarify. How exactly does this comparison support the main conclusions of this manuscript, which involve extremely small spatial scales within a small period of time that would not be well captured by OMI?

We thought that it would be helpful to include satellite measurements, since we compare measurement methods on different scales. Point measurements from the monitoring stations and mobile measurements are combined with path averaging LP DOAs measurements, and satellite measurements provide area or volume averaging concentrations, which are more representative for a larger area, but also include vertical

profiles. We agree that it is not really important for the rest of the study, so according to both reviewers' suggestion, we decided to remove it from the paper, but provide the OMI comparison as supplementary information.

(6) Main result: In my opinion, one of the main results of this manuscript is how many exceedances of the WHO guideline would actually be inferred with a more dense monitoring strategy, compared to the sparse observations currently available. But this result gets glossed over in the rest of the discussion and concluding remarks. I'd be curious if the authors agree that this one of the most important results, and if they think it deserves more prominent position in the abstract/conclusion.

We agree with the reviewer that it is one of the important results. However, we don't want to focus on the number of exceedances of the 1-hour average WHO threshold, because for the comparison we include a lot of assumptions, including the assumption that the drive-by measurement can represent the hourly average. The guideline reference value is not designed for on-road measurements. We think that the direct comparison between on-road concentrations and concentrations sampled at the monitoring stations is more representable.

The specific comments below from the reviewer also have been addressed and corrected in the manuscript. Specific comments:

Introduction:

Page 2, Line 11: "NO2 levels are often strongly correlated with many other toxic air pollutants". Perhaps this is indeed well-known, but I think this statement merits some citations.

Relevant publications have been added as references.

Page 2, Line 25: "measures" should be "measure"

Page 2, Line 25: The authors refer to "defined rules" in terms of measurement height and distance in several places throughout the manuscript, but have mentioned what

these rules are anywhere. I assume these "rules" may vary by jurisdiction. Are there consistent, e.g. European-wide, rules? Does the WHO provide location guidelines? I think at least providing the "rule" for the local agency would be insightful to share somewhere. Even better, share the rules for other places if they vary.

The European directive 2008/50/EC lists a number of criteria for microscale positioning of air quality measurements (Annex III section C). This has been added in Introduction, line 25-30.

Page 3, Lines 15-20: It seems to me that certain portions of the introduction, including these lines, would be better placed in the Methods section of this manuscript.

Since we decided to move OMI section to Supplementary, we keep the demonstration of our purpose there.

Methods:

Page 5, Line 13: The names of each urban station means very little to someone unfamiliar with the region. I later discovered these locations are indeed on the map in Figure 3c. I suggest the authors mention here that these locations can be found on this map.

As reviewer suggested, description added.

Page 6, Lines 4-11: The authors explain that the OMI VCDs are gridded onto a 0.02 x 0.02 grid following approaches from previous work, and later in the manuscript it looks like the authors use monthly averages. Can the authors be slightly more clear here? Are the observations from each day gridded separately, then I assume simple monthly averages are calculated from these daily surfaces?

Since OMI only has one measurement per day pass over Germany, and it is able to detect cloud radiance fraction, cloud pressure and albedo, too. Based on different weather condition, the viewing pixel may be missing and has large uncertainty. Each day is gridded separately and then averaged for one month. The description is added

in Supplementary Section 2 line 10-15.

Page 6, Lines 13-20: Can the authors be clear about what model profiles are being used? I.e., what time of day, and I assume from the same month/year as the satellite observations? Is the surface estimate calculated each day, prior to calculating the monthly average? Or is the surface estimate performed on the monthly average columns? I believe the authors answered these questions in the response to the quick review, but why not include these details in the text for the benefit of readers? Regarding the conversion to surface concentrations, can the authors describe the logic behind using spatially interpolated profiles across four grid cells? Doesn't this represent interpolating over an area of 4x5 degrees? I am curious why they would not simply use the spatially coincident model grid cell.

We decided to move OMI section to Supplementary.

Page 6, Lines 21-22: I'm not sure I agree entirely here. What if the diurnal pattern of sources or chemistry is different over the area of the satellite pixel, compared to the area covered by the LP-DOAS absorption path. Wouldn't that cause a mismatch between the diurnal cycle integrated over each?

That's a good point, we replaced "A good correlation would allow the assumption that the relative diurnal cycle obtained from the LP DOAS can be regarded as being representative for the entire urban area." with "This correlation refers to the spatial coverage of course, and not necessarily also to the diurnal cycle, since OMI measures only once per day."

Results and Discussion:

Page 7, Line 11: "as a general indicator". An indicator of what?

The indicator of city-wide distribution of air quality.

Page 7, Line 13: Why have the concentrations been averaged over the three locations in this case, which presumably represent quite different micro environments? Why not

just report the average concentration of each separately, and within each of their 10-km radius? Section 3.1.1: As I mention above, this really strikes me as belonging in the Methods section. This means you could also keep Figure 2 before Figure 3. As it is now, you reference Figure 3 in the text before discussing Figure 2.

Answered in General comments (2).

Figure 3c: You could be very clear here that this is a zoomed-in plot of the diurnally normalized data.

Yes. It is now clearly indicated in the plot caption.

Table 1: You could be very clear here that the WHO AQG is according to the 24-hr average guideline.

The caption of Table 2 in the paper is modified to WHO AQG 1-hour guideline.

Page 10, Line 17: This is assuming that the WHO guidelines are based on the same monitoring "rules" that are applied by the environmental reporting agency. Is this true? What are the WHO guidelines for monitor placement? Are these the same rules employed by the local air quality agency? Please share these rules explicitly for the benefit of the readers.

Here, according to the European directive 2008/50/EC, a number of criteria for microscale positioning of air quality measurements (Annex III section C) are introduced. This has been added in Introduction, line 25-30.
* * *
[Figure]

**Fig. 1.**

**Table 1.** Overview of the measurement days, time and number of measurements.

| Meas. Date | Meas. Time (hours) | Number of Meas. | Meas. Date | Meas. Time (hours) | Number of Meas. |
|---|---|---|---|---|---|
| 2016.03.21 | 3.84 | 5313 | 2016.06.16 | 5.48 | 7586 |
| 2016.03.22 | 1.82 | 2523 | 2016.06.17 | 2.75 | 3804 |
| 2016.05.09 | 2.29 | 3172 | 2016.07.01 | 2.86 | 3955 |
| 2016.05.10 | 3.62 | 5019 | 2016.07.06 | 6.24 | 17292 |
| 2016.05.11 | 4.55 | 6295 | 2016.07.07 | 10.29 | 28489 |
| 2016.06.06 | 3.38 | 4674 | 2016.07.08 | 5.22 | 14466 |
| 2016.06.07 | 4.66 | 6454 | 2016.07.11 | 5.1 | 14123 |
| 2016.06.13 | 4.25 | 5882 | 2016.07.13 | 2.48 | 2976 |
| Total | 64.8 | 126440 | | | |

**Fig. 2.**

---

## Author Comment (AC2) · 21 Jul 2020

We sincerely thank the reviewer for providing her/his very helpful suggestions, which helped us to improve our paper a lot. Below please find our responses:

1) The mobile data set is interesting. However other mobile data sets examining spatial variability in cities exist and it is not clear if any of the observations here are surprising because they are not placed in the context of the prior related literature. I recommend adding a deeper and more comprehensive discussion of the theoretical and observational understanding we have of emissions from roadways and the length scales of decay of those emissions in cities. For example, papers by Choi et al. including

[Figure]

https://doi.org/10.1016/j.atmosenv.2012.07.084 and Atmos. Chem. Phys., 14, 6925–6940, 2014 and by Apte et al. Environ. Sci. Technol. 2017, 51, 12, 6999–7008 show a characteristic decay length scale of 500-1000m (1/e) that would be approximately consistent with the measurements reported herein. The Choi et al papers also provide a theoretical basis for discussion of the decay. It is also important to note that the time scale for conversion of NO to NO2 is not instantaneous. Thus on-road measurements of NO2 may have a systematic bias. The measurements in Apte, et al. show the consequences of NO to NO2 conversion as different timescales/lengthscales for decay from urban roadway sources.

Thanks very much for reviewer's suggestions. All publications above have been added as references in Section 3.2 line 15-20. These are valuable information to support our research.

2) I find the discussion of the remote sensing measurements confusing. The logic connecting them to the mobile measurements is unclear. It is well-known that OMI measurements with a 2 degree a priori will have a large bias compared to urban measurements. The large context of the long path measurements connects both emissions and loss, while the mobile measurements are so near to the source that they only reflect emissions. I recommend these sections be removed or the connection to the mobile observations made substantially clearer. Also, to help with the readability of the paper, I recommend moving all descriptions of the instruments to the supplement.

According to the reviewer's suggestion, we decided to remove the whole section of OMI comparison from the paper and move it to the supplement.

The description of the instruments is kept in the paper, because we think it is important to describe the different measurement techniques for providing a better understanding for the combination of point and path averaging measurements.